# Template-directed vertical photopolymerization for construction of triphenylamine-based poly(diacetylene) nanofibers

Yingbo Lu[1,2,3], Luyao Jin[1,2], Jiani Wang[1,2], Qiang Fang[4], Shuping Wang[1,2], Jianying Huang [4], Zibin Zhang [1,2] ✉, Feihe Huang [5,6,7] & Shijun Li [1,2] ✉

Template-directed synthesis of macromolecules prevails in natural systems. However, artificial template-directed covalent polymerization that proceeds without sacrificing the delicate non-covalent order needed for precursor alignment remains a formidable challenge. Here we report a supramolecular-templating strategy for photopolymerization of triphenylamine-based diyne assemblies. Cooperative hydrogen- and halogen-bonding align $C_3$-symmetric monomers into ordered stacks that evolve from nanodots into micron-scale nanofibers. Ultraviolet irradiation then triggers axial cross-linking of the diyne moieties, producing continuous one-dimensional conjugated polymers. Selective acid treatment cleaves the I···N halogen bond to remove the template while preserving nanofibrillar integrity, yielding a stable covalent network with red-shifted emission. We demonstrate that this self-assemble-then-cure strategy integrates reversible supramolecular organization with irreversible covalent fixation, providing a general and scalable route to vertically oriented conjugated polymer architectures.

Template-directed synthesis is Nature's universal strategy for building structurally precise biomacromolecules[1,2]. In DNA replication, complementary hydrogen bonds (HB) pre-align nucleobases so that phosphodiester coupling proceeds with single-base fidelity[3,4]. Ribosomal peptide elongation relies on an mRNA "track" whose codons organize amino-acyl-tRNAs through a weave of H-bonding, π-π stacking and $Mg^{2+}$ coordination[5]. Even hierarchical protein folding is guided by networks of HB, salt bridges and hydrophobic contacts that

template covalent disulfide formation[6–8]. Inspired by these biological designs, chemists have spent the past three decades developing artificial, non-covalent templates to drive bond construction with similar precision[9]. HB[10–13], metal–ligand coordination[14–16] and host–guest recognition[17,18] have already enabled the efficient synthesis of discrete but also topologically complex molecules—including crown ethers[19,20], rotaxanes[18,21,22], catenanes[14] and molecular knots[23,24]—whose formation would otherwise require lengthy step-wise routes.

[1]Key Laboratory of Organosilicon Chemistry and Material Technology of Ministry of Education, College of Material, Chemistry and Chemical Engineering, Hangzhou Normal University, Hangzhou, P. R. China. [2]Zhejiang Key Laboratory of Organosilicon Material Technology, College of Material, Chemistry and Chemical Engineering, Hangzhou Normal University, Hangzhou, P. R. China. [3]College of Chemistry and Chemical Engineering, Central South University, Changsha, P. R. China. [4]College of Food Science and Biotechnology, Zhejiang Gongshang University, Hangzhou, P. R. China. [5]State Key Laboratory of Soil Pollution Control and Safety, Stoddart Institute of Molecular Science, Department of Chemistry, Zhejiang University, Hangzhou, P. R. China. [6]Key Laboratory of High-Performance Adhesion Functional Materials and Application Technology of Zhejiang, ZJU-Hangzhou Global Scientific and Technological Innovation Center, Zhejiang University, Hangzhou, P. R. China. [7]Zhejiang-Israel Joint Laboratory of Self-Assembling Functional Materials, ZJU-Hangzhou Global Scientific and Technological Innovation Center, Zhejiang University, Hangzhou, P. R. China. ✉e-mail: zzhang@hznu.edu.cn; l_shijun@hznu.edu.cn

Extending this success to polymer chemistry, however, remains a formidable challenge. Unlike a macrocycle or a mechanically inter-locked pair, a polymer must grow in a directional, repetitive approach while the templating interactions themselves survive the very conditions that drive covalent bond formation. Maintaining a delicate balance between "strong enough to align, yet weak enough not to impede reaction" is therefore the central dilemma in template-controlled polymerization.

Triphenylamine (TPA), comprising a central nitrogen atom connected to three phenyl rings, is a prototypical electron-donating unit with excellent hole-transport capability[25,26]. Since the late 1990s, extensive structural modifications have been undertaken to enhance its key performance parameters like electrical conductivity, optical response, or thermal stability, for organic optoelectronic applications[25,27,28]. Fine control over the polymerization mode of TPA allows systematic tuning of molecular packing, band structure, and solid-state properties, thereby offering fresh strategies for the fabrication of efficient and stable optoelectronic devices.

Lateral covalent polymerization, the most common approach, is to link TPA moieties laterally through covalent bonds. For instance, phenylene–phenylene couplings, alkyne bridges, vinylene units, or N-containing linkers, to afford linear[29,30] or branched[31] conjugated polymers. Extending the π-conjugation in the main chain markedly improves charge transport along the polymer backbone. Nevertheless, device performance remains highly sensitive to π–π stacking and side-chain organization. Defects in intra- or intermolecular packing readily suppress carrier mobility and thus limit overall photovoltaic or current-injection efficiencies.

Vertical polymerization via non-covalent interactions, alternatively, vertical stacking driven by weak forces like HB, van der Waals contacts, or π–π interactions can build supramolecular columns or layers of TPA units[32–35]. Such assemblies are readily reversible and can be tuned by external stimuli, such as solvent, temperature and pH, providing morphological adaptability on the macroscopic scale. In optoelectronic devices, non-covalent vertical stacking facilitates uni- or multidimensional charge-transport channels and may reduce exciton recombination[36,37]. However, the absence of covalent backbones renders these materials susceptible to environmental perturbations; their thermal and chemical stabilities fall short of covalent polymers, and delicate processing is required to preserve the ordered stacking motif[38].

Vertical covalent polymerization, achieving covalent bonding along the stacking (c) axis, would merge the advantages of robust covalent networks with efficient through-plane charge transport[39]. Properly engineered, TPA monomers could form highly ordered one-, two-, or even three-dimensional covalent frameworks featuring periodic pores or interlayer spacings amenable to further functionalization, which is attractive for photocatalysis[26,40], energy storage[41,42], or chemical sensing[43]. Nonetheless, how to actualize vertical covalent polymerization is a great challenge. First, formation of axial bonds typically demands high-energy inputs (light, heat, reactive catalysts) that can weaken or disrupt the fragile non-covalent interactions needed for pre-organization. Balancing a "driving force sufficient for bond formation" against "mild conditions that preserve stacking" represents a major synthetic dilemma. Second, guiding principles for such axial polymerization remain poorly defined. Reported successes frequently rely on crystal-engineering strategies, namely pre-crystallization followed by topochemical photopolymerization[44,45] or clamping templates[46], both of which suffer from low yields and sensitivity to substituents and charge distribution. Scalable fabrication of thin films or coatings, therefore, calls for a dual-control methodology in which *self-assembly* and *bond formation* are both precisely regulated[47]. A promising solution lies in combining reversible non-covalent motifs with photo-induced cross-linking, enabling "in-situ curing" once the initial alignment has been achieved.

In this work, we present a supramolecular-templating concept that employs cooperative HB and halogen-bond (XB) interactions to pre-align $C_3$-symmetric molecules, subsequently exploiting this alignment in a photochemical polymerization reaction. This design strategy takes distinctive advantages of supramolecular assembly as a template for polymerization, which offers a perspective for the rational manipulation of matter at the molecular scale.

## Results

Figure 1 and Supplementary Fig. 1 presents the $^1$H NMR spectra of **1**, **2**, and a series of **1** + **2** mixtures recorded in $CD_2Cl_2$. Compared to pure **2** (Fig. 1a), addition of three equivalents of **1** produces a slight up-field shift of the pyridyl-H (H$^3$) resonance (Fig. 1b), signifying the formation of an incipient I···N XB between the σ-hole of the perfluoro-iodoaryl unit in **1** and the pyridyl nitrogen in **2**. When the concentration of **2** is raised stepwise to 20 mM (Fig. 1b → 1f), this shift becomes progressively larger, this trend reflects an increase in the proportion of supramolecular assemblies formed at higher concentrations, as a greater fraction of the monomers are engaged in HB/XB-mediated aggregates. In a control experiment, the amide groups of **1** were replaced by esters to give compound **3**; a 3:1 (molar ratio) mixture of **3** + **2** with identical concentration (Fig. 1g) produces virtually no chemical-shift change in H$^3$, demonstrating that an isolated XB is too weak in this system and must be reinforced by amide-based HB pre-organization.

Concomitantly, the amide-NH protons (H$^1$ and H$^2$) of **1** display a pronounced down-field shift upon increasing concentration (Fig. 1b–f), indicating that a greater proportion of the molecules are engaged in HB/XB-mediated supramolecular assemblies at higher concentrations. Comparison of the most concentrated **1** + **2** sample (Fig. 1f) with pure **1** of the same concentration (Fig. 1h) shows a further down-field displacement of H$^1$ and H$^2$, confirming that the XB stabilizes and reinforces the HB network. In contrast, the aromatic proton H$^4$ on compound **1** exhibits only a minor chemical-shift change upon mixing with compound **2**, and this shift shows no systematic dependence on concentration (Fig. 1b–f). This behavior indicates that H$^4$ is not directly involved in the primary HB/XB recognition motif. Instead, the slight displacement of H$^4$ is attributed to a secondary electronic or magnetic environment effect induced by aggregate formation, rather than a specific directional interaction. The absence of a concentration-dependent shift further suggests that the supramolecular assembly is dominated by the amide hydrogen bonds and the I···N halogen bond, while the phenyl ring bearing H$^4$ plays a largely spectator role in the assembly process.

Gelation test also supports the NMR data: a 1.5 mM solution of **1** in methylcyclohexane (MCH) remains a free-flowing liquid (Supplementary Fig. 2a), whereas a mixture of 1.5 mM solution of **1** and 0.5 mM of **2** transfers to a non-flowing gel (Supplementary Fig. 2b). The gel undergoes a complete gel-to-sol transition within 1 minute upon contact with HCl vapor, accompanied by a color change to deep yellow (Supplementary Fig. 2c), which implying acid-triggered disassembly of the ordered aggregate created by cooperative HB/XB interactions.

To gain deeper insight into their self-assembly behavior, we determined the binding stoichiometry between compound **1** and a pyridyl-bearing acceptor by fluorescence titration (Fig. 2a). Because the diyne moiety is highly sensitive to UV irradiation, direct titration of **2** was not feasible. We therefore selected compound **4**, which contains the same pyridyl recognition site but lacks the diyne unit, as a surrogate probe. Incremental addition of **1** to a MCH solution of **4** resulted in gradual fluorescence quenching, and the corresponding mole-ratio plot (Fig. 2b) displayed a clear inflection at a 3:1 ratio, indicating the formation of a stable complex comprising three molecules of **1** and one of **4**. Given that **2** and **4** possess identical pyridyl binding sites and share the same triphenylamine core, we infer that the binding stoichiometry between **1** and **2** under the same conditions should also be 3:1.

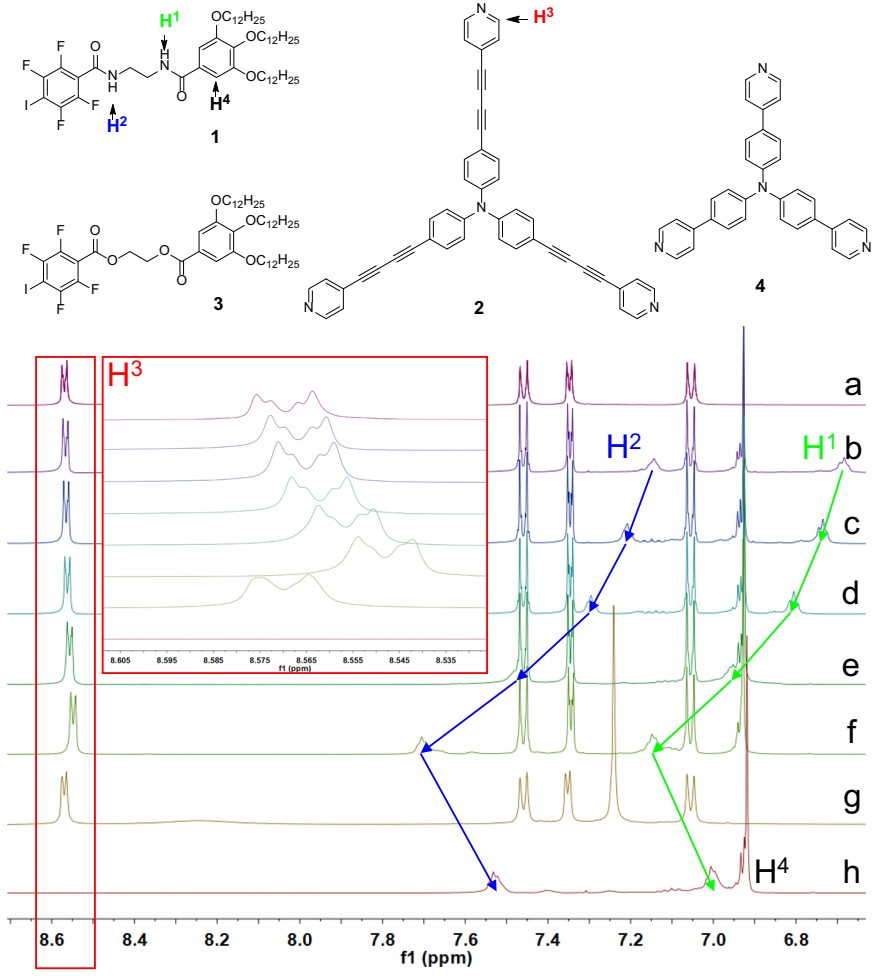

**Fig. 1 | The chemical structures of 1, 2, 3, 4 and characterization of their self-assemblies.** The partial ¹H NMR (500 MHz, CD₂Cl₂, 298 K) spectra of (**a**) **2** (20 mM); (**b**) **1** (3 mM) + **2** (1 mM); (**c**) **1** (7.5 mM) + **2** (2.5 mM); (**d**) **1** (15 mM) + **2** (5 mM); (**e**) **1** (30 mM) + **2** (10 mM); (**f**) **1** (60 mM) + **2** (20 mM); (**g**) **3** (60 mM) + **2** (20 mM); (**h**) **1** (60 mM).

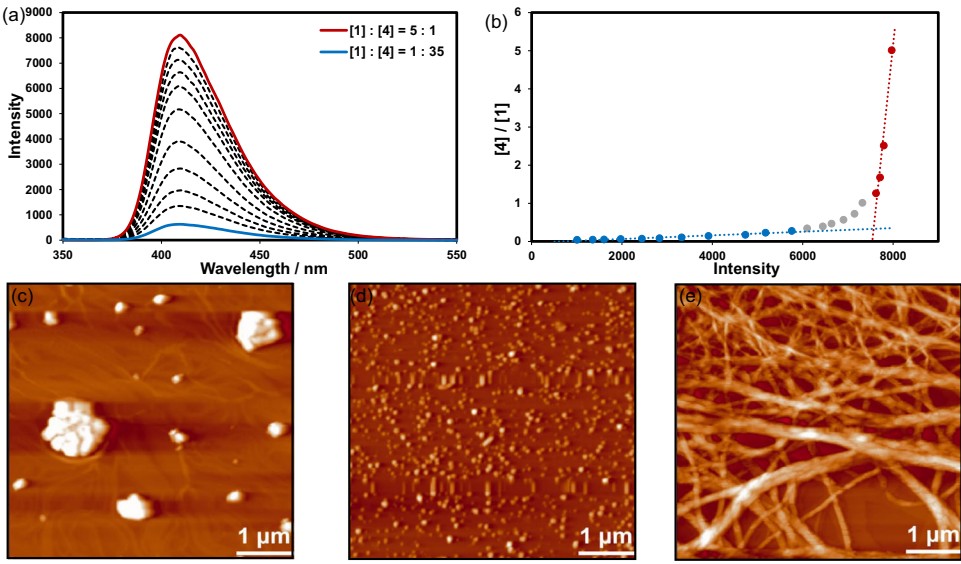

**Fig. 2 | Cooperative supramolecular complexation of 1 and 4, and morphological evolution of 1 and 2. a** The fluorescence spectra of compound **4** (50 μM in MCH) upon addition of **1** at 25 °C, and (**b**) the mole ratio plot for the complexation between **1** and **4**, indicating a 3:1 stoichiometry; and the AFM images of (**c**) **1**, (**d**) **2** and (**e**) **1** + **2**.

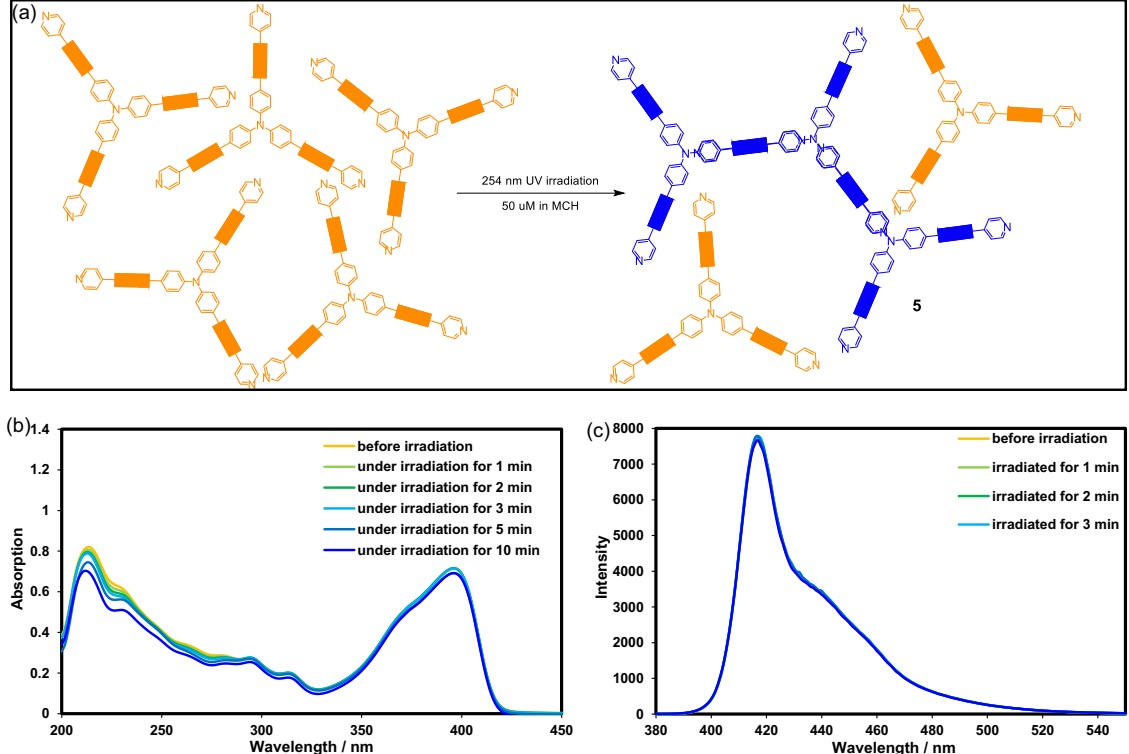

**Fig. 3 | Photo-crosslinking behavior of 2 in the absence of 1. a** The photo-crosslinking reaction of **2** (50 μM) under UV (254 nm, 3 W) in MCH, and the (**b**) UV-vis and (**c**) FL spectra during 10 min of irradiation.

AFM imaging provides morphological evidence for this cooperative pre-organization (Fig. 2c–e). Pure **1** deposits as irregular plate-like clusters with diameters of 100–300 nm, consistent with face-to-face aggregation driven solely by bifurcated HB; the intrinsic asymmetry of **1** offers no axial guidance, so no long-range one-dimensional structures are observed. Pure **2** yields only uniformly dispersed nanodots (< 10 nm), indicating that it remains molecularly dispersed or forms very small aggregates upon solvent evaporation. In stark contrast, a 3:1 molar mixture of **1** and **2** produces a dense mat of interwoven nanofibers with several micrometers in length, eventually developing into a three-dimensional network. The transformation from plates (**1**) or dots (**2**) to high-aspect-ratio fibers (**1** + **2**) demonstrates that HB alone generates disordered aggregation, whereas the introduction of **2** supplies XB that, together with HB, "zipper" multiple **1** and **2** units into axially aligned stacks, inhibiting side-chain motility and driving parallel arrangement of the diyne moieties. Transmission electron microscopy (TEM) further corroborates the cooperative self-assembly behavior inferred from AFM. As shown in Supplementary Fig. 3, pure compound **1** does not form ordered nanofibrous structures. In contrast, the 3:1 mixture of **1** and **2** gives rise to well-defined nanofibers with high aspect ratios, extending from hundreds of nanometers to micrometer lengths. The pronounced morphological difference between the mixed system and the individual component confirms that the cooperative action of hydrogen bonding and halogen bonding is crucial for directing axial growth and templating the supramolecular framework prior to photopolymerization. During solvent removal, these stacks extend over the substrate to give continuous, highly oriented one-dimensional nanofibers. This transition to a more ordered hierarchical morphology provides a well-defined pathway for subsequent photochemical cross-linking and, together with the NMR results, unequivocally substantiates the crucial role of cooperative XB and HB in enabling controlled, ordered polymerization.

When compound **2** (50 μM, MCH) is irradiated without presence of **1** (Fig. 3a), its UV spectrum (Fig. 3b) shows a gradual attenuation of the intense π–π band centered at 220 nm, whereas the 270–420 nm region is essentially unchanged. Because this 220 nm transition derives mainly from the 1,3-butadiyne chromophore, the intensity loss indicates partial degradation of the π system, probably via minor, non-directional butadiyne cross-linking. No isosbestic point or band appears, confirming the absence of a well-defined, selective transformation. In solution, the butadiyne axes of **2** are randomly oriented, and therefore precluding efficient topochemical polymerization.

In contrast, the 3:1 mixture of **1** and **2** exhibits diagnostic spectral changes (Fig. 4b). During the first 0–3 min the 220–240 nm π–π manifold decays rapidly and a pronounced isosbestic point emerges at 266 nm, indicating a clean two-state conversion. Simultaneously, a broad low-energy band grows continuously from about 280 to 390 nm. The latter is characteristic of the "blue phase" poly(diacetylene) generated by 2π + 2π cross-linking of butadiyne units positioned at reactive distances.

Fluorescence spectroscopy corroborates these findings (Figs. 3c, 4c). Compound **2** itself shows a stable emission maximum at 417 nm. After 10 min irradiation, the intensity drops by 2%, demonstrating negligible chemical conversion and the absence of ACQ/AIE effects. By contrast, the irradiation of a **1** + **2** mixture causes a mono-exponential decrease in emission intensity while retaining the same $\lambda_{max}$ (418 nm). No band or discernible red shift appears, indicating that the existing chromophore is progressively "darkened" because of the strengthened π–π interaction of TPA moieties.

AFM imaging (Fig. 4d-f) visualizes the morphology of the 3:1 (150 μM **1** + 50 μM **2**) system at successive irradiation times. After radiated for 2 min, predominantly flexible nanofibrils (25–35 nm diameter) with large bending radii are observed, indicating that only a fraction of butadiynes have reacted. Following 5 min of UV exposure, fiber density increases markedly; inter-fiber entanglement yields a dense felt-like network and individual strands thicken to 30–40 nm, reflecting widespread axial cross-linking and enhanced chain rigidity. After being exposed to UV light for 10 min, the network contracts

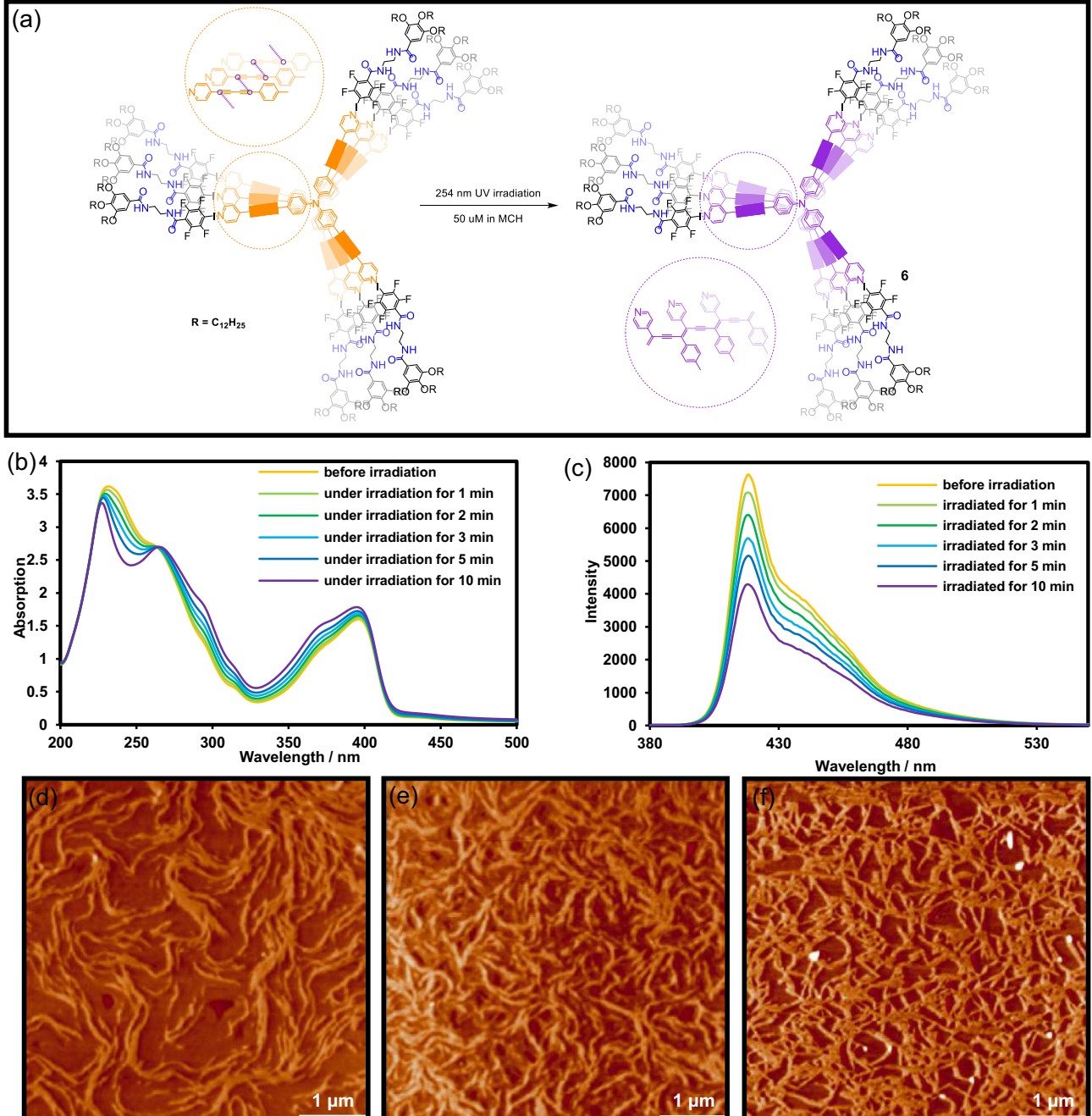

**Fig. 4 | Photo-crosslinking behavior of 2 in the presence of 1. a** The photo-crosslinking reaction of **2** (50 µM) in existence of 3 equiv of **1** under UV (254 nm, 3 W) in MCH, the (**b**) UV-vis, (**c**) FL spectra during 10 min of irradiation, and the AFM images under irradiation for (**d**) 2 min; (**e**) 5 min; (**f**) 10 min.

further and fuses at junctions, ultimately forming a rigid mesh of uniform pores with beam thicknesses of 40–60 nm.

This "soft bundle - interwoven felt - hardened mesh" progression mirrors the spectroscopic evolution: decay of the high-energy absorption, increase of the 400 nm poly(diacetylene) band, and first-order fluorescence quenching. Together with NMR evidence, the AFM results confirm that cooperative XB and HB templating creates ordered supramolecular channels that support controlled, one-dimensional polymer growth.

Because compound **2** contains three pyridyl groups, it can form pyridinium hydrochloride salts upon treatment with HCl, rendering it soluble in water. In contrast, compound **1** possesses three dodecyl chains and is therefore more soluble in organic solvents (MCH). Taking advantage of the acid sensitivity of the I···N XB, we added HCl solution

(1 M) to the reaction mixture to selectively dissociate and remove compound **1** from the **1** + **2** assembly. The aqueous layer was then collected, and saturated NaOH solution was added to remove the acids from the protonated pyridinium salts of **7**. After basification, the resulting polymer precipitated from the aqueous phase. Dichloromethane was subsequently added to extract the polymer **8** from the aqueous suspension (Fig. 5a). The purified axial polymer **8** was characterized by [1]H NMR, FTIR, UV–vis, FL, AFM, and benchmarked against the product obtained by photo-crosslinking **2** alone under identical conditions (Fig. 5).

The two products show pronounced different [1]H NMR spectra, including chemical shifts, line shape and half-height line width (Fig. 5b). After irradiated for 10 min at 254 nm (blue trace), compound **2** itself still displays sharp pyridyl resonances at 8.61 and 7.36 ppm, and

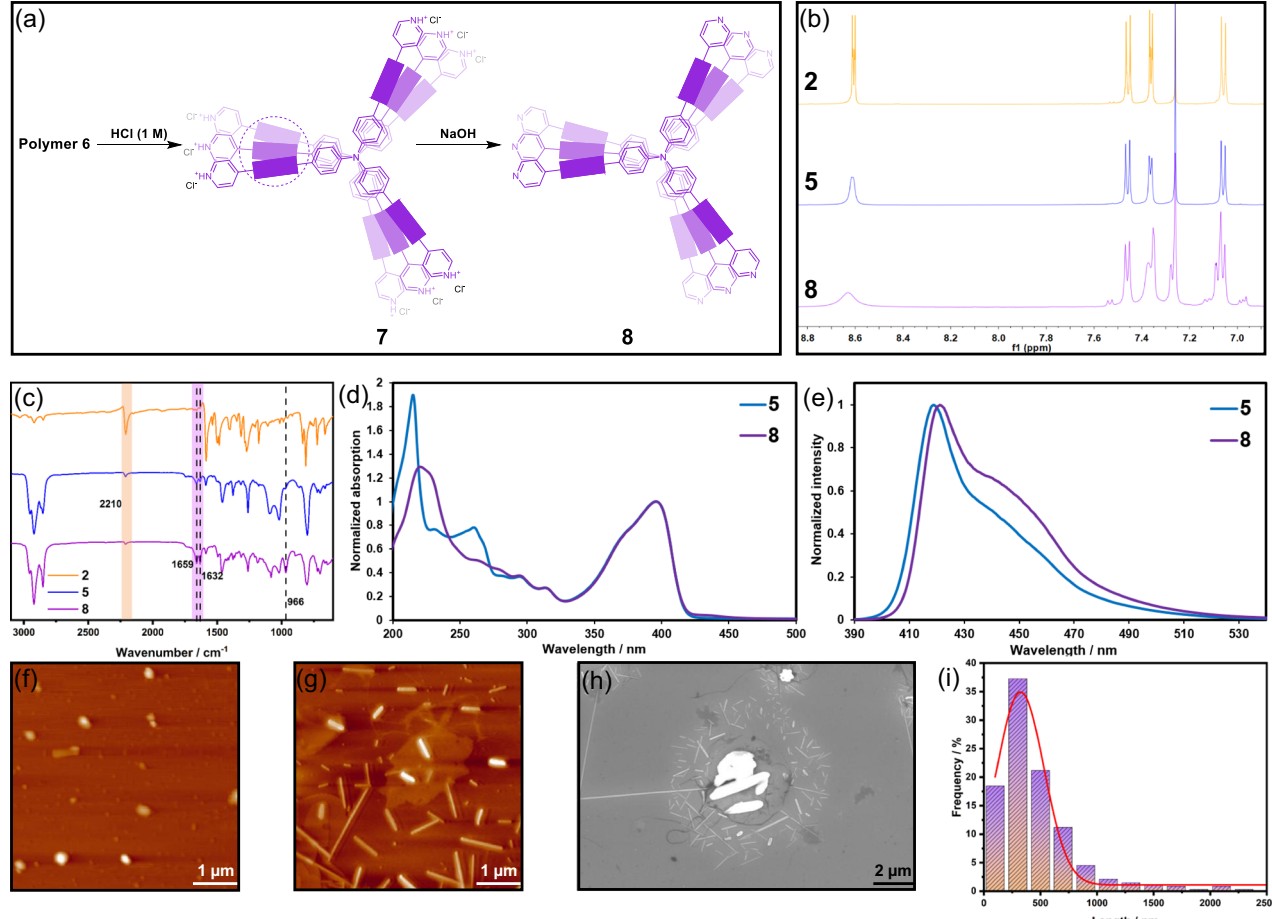

**Fig. 5 | Template removal and structural characterization of the axially polymerized product. a** Schematic illustration of the acid/base work-up process for removing the supramolecular template **1** from polymer 6 to afford the freestanding PDA polymer **8**; (**b**) ¹H NMR (500 MHz, CDCl₃, 298 K) and (**c**) FTIR spectra of **2**, **5** and **8**; (**d**) UV-vis and (**e**) FL spectra of **5** and **8**; AFM images of (**f**) **5** and (**g**) **8**; (**h**) SEM image of **8** and (**i**) length distribution histogram of nanofibers obtained from statistical analysis of several hundred individual objects, fitted with a Gaussian profile.

well-resolved TPA multiplets at 7.05–7.47 ppm, indicating that only local, non-directional cross-linking has occurred, the backbone remains flexible and molecular tumbling is fast. In striking contrast, the spectrum of the template-guided polymer **8** (purple trace) contains no resolvable sharp peaks at 8.61 ppm. Instead, a broad peak with diffuse shoulders is observed at 8.63 and 7.37 ppm. Severe broadening of the pyridyl protons and swelling of the aromatic region imply a highly cross-linked, rigid environment in which both lateral and axial motions are strongly restricted. Notably, resonances attributable to the aromatic protons of compound **1** are not discernible in the spectrum of polymer **8**. Given the strong broadening associated with the rigid poly(diacetylene) (PDA) backbone, any residual signals from **1** would be expected to appear in the aromatic region (Fig. 1 and Supplementary Fig. 7, 6.93 ppm) if present in significant amounts. Their absence therefore indicates that the supramolecular template has been effectively removed through the acid/base work-up, supporting the feasibility of the stimulus-triggered template removal strategy.

FTIR analysis provides molecular-level insights into the photochemical outcomes with and without supramolecular templating (Fig. 5c). For both of the non-templated product **5** and the templated photoproduct **8**, the C≡C band of the diacetylene precursor at 2210 cm⁻¹ is significantly attenuated relative to the monomer, indicating that photopolymerization of **2** proceeds under UV irradiation in both cases. However, clear distinctions emerge in the subsequent evolution of the conjugated backbone. The absorptions at 1659 and 1632 cm⁻¹, which are characteristic of PDA-type conjugated C=C

stretching, are markedly more intense in **8** than in **5**, suggesting that the presence of the HB/XB supramolecular template promotes a more efficient and structurally coherent topochemical polymerization process. Moreover, a distinct band at 966 cm⁻¹ is observed in **8** and is assigned to the out-of-plane =C–H bending vibration of *trans*-vinylene (–CH=CH–) units embedded in the PDA backbone[48]. In contrast, this feature is weak or barely detectable in **5**. These vibrational signatures collectively indicate that, although photochemical coupling of **2** occurs in the absence of templating, only the templated system supports the formation of a more extended and conformationally uniform PDA backbone. Semi-quantitative analysis of the C≡C band confirms this conclusion: the residual diyne signal after irradiation corresponds to an estimated conversion of 96% in the templated system. Although this value is derived without internal normalization and should therefore be interpreted as semi-quantitative, it strongly supports the assertion that the vast majority of diacetylene groups undergo topochemical coupling when guided by the HB/XB supramolecular template.

Fluorescence spectroscopy further differentiates the two photoproducts (Fig. 5e). Compound **2** itself cross-links only in a random, freely diffusing environment, its longest-wavelength emission maximum remains at 419 nm after irradiation. In comparison, owing to the preorganisation through cooperative amide HB and I···N XB, the **1** + **2** assembly undergoes oriented cross-linking of the butadiyne axes, resulting in a long-range conjugated poly(diacetylene). After removal of **1**, polymer **8** emits at 421 nm, that is, red-shifted by 2 nm, and shows significant band broadening with an elevated tail from 430 to 470 nm.

The red shift, accompanied by tail growth, is attributed to chain-extension of the π-system and inter-chain J-type excitonic coupling, which jointly lower the π–π transition energy without altering the chromophoric core.

AFM imaging provides direct structural evidence for the divergent reaction pathways. In the absence of a template (Fig. 5f), the product of **2** (polymer **5**) appears as randomly distributed spherical particles. Occasional bright aggregates are assigned to locally collapsed amorphous domains. The data confirm that, without pre-organization, photo-crosslinking of **2** is confined to isolated molecules or loose clusters and cannot yield long, one-dimensional conjugated segments. In sharp contrast, the material obtained by using the **1** + **2** templating system after photolysis and template removal retains rod-like (Fig. 5g) nanofibers, demonstrating that the axially polymerized poly(-diacetylene) backbone survives the acidic and basic work-up.

To further quantify the axial extent of the covalent framework in the final polymer, statistical length analysis was performed on several hundred nanofibers based on SEM images (Fig. 5h, i). The resulting length distribution follows a Gaussian profile with a most probable length of approximately 324 nm. Using the reported repeat distance of the poly(diacetylene) backbone (0.49 nm)[49], this characteristic length corresponds to an apparent axial degree of polymerization on the order of 600–700 repeating units. Although these nanofibers do not necessarily represent individual polymer chains, the length statistics provide quantitative evidence that cooperative HB/XB templating enables the formation of extended, continuous covalent backbones over hundreds of nanometers, substantially exceeding the conjugation lengths typically attainable through random solution cross-linking. Importantly, the covalent core remains an uninterrupted one-dimensional conduit. This also confirms the success of our strategy to synthesize vertically polymerized structures through the cooperative effect of HB and XB.

## Discussion

By harnessing mutually reinforcing HB and XB, we achieved precise axial pre-alignment of diyne-conjugated TPA chromophores and converted this order into a continuous covalent backbone under 254 nm UV irradiation. Comprehensive spectroscopic and microscopic analyses confirm that HB/XB cooperativity is essential for forming high-aspect-ratio supramolecular channels, and the pre-organized lattice supports photo-polymerization. Finally, the sacrificial supramolecular template can be quantitatively removed to expose an uninterrupted poly(-diacetylene) core whose mechanical rigidity and electronic conjugation exceed those attainable through random solution cross-linking. The study not only validates a "self-assemble-then-cure" paradigm for vertical covalent polymerization but also offers a scalable strategy for constructing vertically aligned polymers. Beyond the triphenylamine system explored here, preliminary supramolecular screening suggests that the HB/XB-templated vertical pre-organization is not limited to a single molecular framework and may be extendable to other pyridyl-containing symmetric monomers. Although a full validation of molecular scope lies outside the focus of this study, these observations indicate that the present "self-assemble-then-cure" strategy has the potential to serve as a general platform for constructing vertically aligned conjugated nanostructures. This conceptual framework, which integrates reversible HB/XB guidance with in-situ photochemical locking, demonstrates strong generality. It is applicable to other directional bond-forming reactions and lays the foundation for the development of next-generation optoelectronic, sensing, and filtration devices requiring long-range molecular precision.

## Methods
### Materials
2,3,5,6-Tetrafluorobenzoic acid, *n*-butyllithium, iodine, methyl 3,4,5-trihydroxybenzoate, potassium carbonate, 1-bromododecane, ethane-1,2-diamine, thionyl chloride, 1,4-bis(trimethylsilyl)buta-1,3-diyne, MeLi·LiBr, 4-iodopyridine, copper(I) iodide, tetrakis(triphenylphosphine)palladium, tetrabutylammonium fluoride, tris(4-iodophenyl)amine, trans-bis(triphenylphosphine)palladium(II) chloride, ethane-1,2-diol, anhydrous $Na_2SO_4$, and all of the solvent used were purchased and used without further purification. The chromatographic separation was carried out on 300–400 mesh silica gel under pressure. $^1H$, $^{13}C$ and $^{19}F$ NMR spectra were recorded on a Bruker AVANCE III-500M NMR spectrometer at room temperature, and chemical shifts are reported in ppm ($\delta$) with the signal of tetramethylsilane (TMS) as internal standard at 0.00 ppm. MS spectra were measured on an Agilent 1290LC-6530QTOF. UV-vis absorption spectra were recorded on a SHIMADZU UV-2600i. Fluorescent spectra were recorded on a HITACHI F-7000. AFM imaging was performed under ambient conditions using a Being Nano-Instruments CSPM5500Z in Tapping mode. Samples were drop-casted onto a silicon wafer which was treated with piranha solution.

### General procedure for sample preparation (UV-vis, FL, AFM, SEM)
Compound **1** (3.06 mg, 0.003 mmol) and compound **2** (0.62 mg, 0.001 mmol) were mixed and dissolved in 1 mL dichloromethane. The solvent was removed under vacuum to afford a uniform residue. Subsequently, 20 mL of MCH was added, and the mixture was sonicated to ensure complete dispersion. The resulting suspension was heated to 40 °C and maintained for 1 min. The sample was then cooled to room temperature and stood for 1 h.

### Procedure for photo-crosslinking reaction
Compound **1** (12.23 mg, 0.012 mmol) and compound **2** (2.48 mg, 0.004 mmol) were mixed and dissolved in toluene (10 mL). The mixture was sonicated for 3 min and then stood overnight.

A UV lamp (254 nm, 3 W) was inserted directly into the reaction vessel and positioned 0.5 cm above the liquid surface. The solution was irradiated for 10 min. Upon completion of irradiation, HCl (1 M, 10 mL) was added, and the mixture was vigorously stirred for 10 min. The aqueous layer was then collected, and this acid-washing procedure was repeated for three times. The combined aqueous layers were then basified by slowly added saturated NaOH solution until the pH > 12. The basic aqueous phase was extracted with dichloromethane (10 mL × 5), and the combined organic extracts were dried over anhydrous $Na_2SO_4$, filtered, and concentrated under reduced pressure to afford polymer **8**.

The photocrosslinking reaction of pure compound **2** was carried out using a similar procedure.

### Procedure for AFM, SEM and TEM sample preparation
The sample produced by the general procedure was drop-casted onto a carbon-coated grid (Cu, 400 mesh) or a silicon wafer, which was treated with piranha solution.

### The treatment of silicon wafer
In an ice–water bath, 10 mL of hydrogen peroxide (30%) was slowly added to 30 mL of sulfuric acid (98%) with gentle stirring using a glass rod. After the exothermic mixture was allowed to cool to room temperature, the silicon wafer was immersed in the solution for 20 min. Subsequently, the wafer was removed, and washed for more than 5 times with deionized water, and finally stored in isopropanol until use.

### Procedure for gel preparation
A gel composed of **1** and **2** was prepared as follows: Compound **1** (1.53 mg, 0.0015 mmol) and compound **2** (0.31 mg, 0.0005 mmol) were mixed and dissolved in 0.5 mL dichloromethane. The solvent was removed under vacuum to afford a uniform residue. Subsequently, 1.0 mL of MCH was added, and the mixture was sonicated to ensure complete dispersion. The resulting suspension was heated to 70 °C

and maintained for 1 min. The sample was then cooled to room temperature and allowed to stand overnight, giving a stable supramolecular gel.

The gel derived from **1** alone was prepared using an identical procedure.

The synthesis and characterization of compounds presented in this work, and additional data of tests are described in the Supplementary Information.

## Data availability

The authors declare that the data supporting the findings of this study are available within the paper and its Supplementary Information file. All other data are available from the corresponding authors upon request. Source data are provided in this paper.

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

## Acknowledgements

S.L. thanks the Natural Science Foundation of Zhejiang Province (LZ24B020005) and S.W. thanks the Natural Science Foundation of Zhejiang Province (LQN25B020009) for financial support.

## Author contributions

S.L. conceived and designed the experiments. Y.L., L.J., and J.W. performed the synthesis of the molecules. Y.L., Q.F., S.W., and J.H. performed the self-assembly experiments, photo-cross linking experiments and UV-vis, FL and AFM studies and analyzed the data. Y.L., Z.Z., F.H., and S.L. wrote the manuscript together.

## Competing interests

The authors declare no competing interests.
