## [Transparent Peer Review file · Nature Communications]

Template-Directed Vertical Photopolymerization for Construction of Triphenylamine-Based Poly(diacetylene) Nanofibers

Corresponding Author: Professor Shijun Li

Version 0:

Reviewer comments:

Reviewer #1

(Remarks to the Author)

In the manuscript titled as "Template-Directed Vertical Photopolymerisation for Construction of Triphenylamine-Based Poly(diacetylene) Nanofibers", Lu and coworkers presented an innovative self-assemble then cross-linking strategy for fabricating vertically aligned conjugated polymers via cooperative hydrogen/halogen bonding and photopolymerization. The concept of integrating supramolecular pre-organization with covalent fixation is promising for nanomaterials design and applications. Historically, template-directed polymerizations that rely on preorganized vertical stacking have been extremely rare, because it is challenging to preserve delicate non-covalent order while forming covalent linkages along the stacking axis. The authors successfully overcome this dilemma, converting reversible supramolecular assemblies into stable one-dimensional covalent polymers under mild photochemical conditions. Therefore, it is a very nice work and appeals to the broad readership of Nature Communications, especially for polymer and supramolecular chemists. I thus think that this manuscript deserves to be published after addressing the issues described below.

1. In this templating polymerization protocol, self-assembly of the precursors driven by hydrogen/halogen bonding is very crucial for the subsequent photopolymerization. It is an important prerequisite for the ordered polymerization. However, in the manuscript, the investigation on the self-assembly behaves is not deep enough. It is recommended to do more studies on the self-assembly before photopolymerization, such as the measurement of binding stoichiometries and other characterizations (e.g. TEM or other electron microscope observations), which may provide more useful information to understand the templating mechanism.

2. The overall English expression of the article is basically accurate, but there are some minor issues as follows. In the title, the authors used "polymerisation", however in the main section, the word "polymerization" appeared several times. I think both of them are OK, but it is better to choose one in a single paper. On page 1, the abstract, the "align C3-symmetric monomers into ordered stacks" should be change into "C3-symmetric". At the end of page 3, "van der Waals contacts, or π - π interactions can builds supramolecular columns" should be "can build".

On page 4, the authors said that "In optoelectronic devices, non-covalent vertical stacking facilitates uni- or multidimensional charge-transport channels and may reduce exciton recombination." It is better to add some related references, making the sentence more convincing.

On page 6, the caption of Fig 1, it should be "The chemical structures of 1, 2, 3." When they discussed the NMR spectra, they said "the pyridyl-H resonance (Fig. 1b)" and "the amide-NH protons of 1". However, these protons already numbered as H3, H2 and H1 as shown in Fig 1. They can use these numbers when discussing them.

On page 9, they said that "without presence of 1 (Fig. 3, middle)," It is not very clear. They should divided Fig 3 into two parts, like Fig 3a and Fig 3b for the preparation of compound 6 and 5, respectively.

3. There are some issues with the following pictures.

Figure 3 can be enlarged somehow. The structures in Figure 3 are too small and not clear enough. The structure of aggregate 4 appeared twice. This is not necessary. It can be combined with the preparation of polymer 6. And place the preparation of 5 in the lower part as Fig 3b.

The AFM image of compound 5 shown in Figure 5d is not clear enough and may cause confusion for the readers. It is

suggested to replace it with a clearer one.

4. Since the characterization of AFM is crucial for understanding the morphology of the assemblies and vertically aligned conjugated polymer 6, I suggest adding a description of the AFM sample preparation in the SI section. This will facilitate researchers in the same field to follow.

Reviewer #2

(Remarks to the Author)

In this manuscript, the authors present a conceptually elegant and technically rigorous study that reports a new synthetic strategy for achieving vertical covalent polymerisation of triphenylamine-based diacetylene assemblies through cooperative hydrogen-bonding (HB) and halogen-bonding (XB) interactions. This “self-assemble-then-cure” approach represents a conceptual advance with far-reaching interdisciplinary implications, providing a new synthetic platform that bridges supramolecular chemistry, polymer science, materials engineering, and optoelectronic applications. The integration of HB/XB cooperativity as a dynamic yet directional template not only achieves vertical polymerisation in solution—beyond crystalline topochemical constraints—but also opens a scalable route to conjugated nanostructures with high structural precision.

Overall, this work is original, methodologically sound, and of clear significance to a broad readership spanning chemistry, physics, and materials science. However, before the manuscript can be accepted for publication, several minor revisions should be made to improve clarity and consistency.

1. Ensure all acronyms (HB, XB, PDA, MCH, etc.) are defined upon their first use in both the main text and supporting information.
2. In Figure 4, specify the UV irradiation power density, wavelength, and apparatus geometry.
3. In Figure 4, the four UV–vis and fluorescence spectra should be properly aligned for visual consistency. The panel labels (a), (b), (c), and (d) are currently misplaced and should be positioned uniformly at the upper left corner of each subfigure to comply with journal formatting standards.
4. In Figure 5, there are several layout inconsistencies that should be corrected in the revised version. Specifically, the border colors of panels (b) and (c) are not identical, and there is a visible white gap between panels (d) and (e). These formatting inconsistencies should be unified for a cleaner and more professional presentation.
5. The procedures for gel preparation, photopolymerisation (photo-crosslinking) experiments, and the treatment after irradiation should be described in greater detail in the Supporting Information. Providing specific information on concentrations, irradiation time and intensity, and purification (e.g., washing or drying steps) would significantly improve the reproducibility and transparency of the experimental methodology.
6. The concept of achieving vertical photopolymerisation by pre-organising C₃-symmetric triphenylamine derivatives is intriguing. However, how can it be a general method and what are the potential applications? At least, the authors should discuss the potential generality and potential applications of this method in the manuscript, which would broaden its impact and inspire future studies in related fields.

Reviewer #3

(Remarks to the Author)

This manuscript by Zhang, Li, and co-workers reports a supramolecular “self-assemble-then-cure” strategy to obtain triphenylamine-based poly(diacetylene) (PDA) nanofibers. Cooperative amide hydrogen bonding together with I···N halogen bonding pre-organizes a diyne monomer into 1D stacks, and short 254 nm UV irradiation triggers axial 2π+2π photopolymerization. After polymerization, the sacrificial template is removed by acid/base work-up, yet the nanofibrous morphology is retained. Evidence for templated assembly, photoconversion, and post-template integrity is provided by UV–vis/fluorescence spectroscopy, AFM imaging, and comparative ¹H NMR analyses collected before and after templating/polymerization.

The assembly–photopolymerization–template-removal concept is attractive and the qualitative evidence is convincing. However, the characterization of the resulting polymer is not yet at the level expected. Adding (i) at least one molecular-weight/DP handle, (ii) one vibrational or solid-state confirmation of PDA backbone formation/order, and (iii) clearer yield/conversion reporting would upgrade the work from a qualitative demonstration to a quantitatively characterized polymer material suitable for a broad materials audience.

A major weakness is the absence of standard polymer metrics (Mn/Mw/Đ). At present the study establishes that “a polymer is formed,” but it does not show how large or how disperse it is. The authors are encouraged to consider one or more of the following routes:

*Controlled scission to generate soluble fragments, followed by SEC-MALS to obtain Mn/Mw/Đ;

*An end-group/chain-stopper strategy (low-% chromophoric stopper) combined with qNMR or absorbance ratios to estimate DP;

*AFM/TEM length statistics on a large number of fibers, converted to DP using a conservative repeat distance.

In addition, backbone verification by vibrational spectroscopy is missing. The authors should include Raman and/or IR data

diagnostic for diacetylene → PDA conversion ($C\equiv C$ loss, $C=C$ growth), and—if possible—quantify the conversion by band-area analysis to complement UV–vis kinetics. One solid-state method (ssNMR for backbone signals or PXRD for topochemical order) would further support claims of an ordered 1D conjugated structure. It would also help to report elemental analysis (e.g., halogen/N content before and after work-up) and/or provide simple kinetic fitting of the spectral evolution to estimate the degree of conversion/cross-linking and to demonstrate complete template removal.

Overall, the core narrative—from self-assembled axial photopolymerization to template removal—is compelling and already well supported by spectroscopy and AFM. The manuscript will be substantially strengthened by the above quantitative additions, which will move it from “polymer formed” to a fully and rigorously characterized polymer material.

Reviewer #4

(Remarks to the Author)

Version 1:

Reviewer comments:

Reviewer #1

(Remarks to the Author)

In the revised version, all of my suggestions and comments have been fully considered and necessary additional experiments were performed along with the proper discussion. Therefore, this manuscript should be accepted for publication.

Reviewer #2

(Remarks to the Author)

The authors have addressed the issues raised by the reviewer. I recommend the publication of this manuscript in its current form.

Reviewer #3

(Remarks to the Author)

The authors have adequately answered and addressed all the concerns raised in the previous round of review. I recommend the manuscript for publication in its current form.

Reviewer #4

(Remarks to the Author)

We appreciate the reviewers' comments which have greatly improved our manuscript. The specific changes to the manuscript as per request are listed as following.

Response to Reviewer 1:

"In the manuscript titled as "Template-Directed Vertical Photopolymerisation for Construction of Triphenylamine-Based Poly(diacetylene) Nanofibers", Lu and coworkers presented an innovative self-assemble then cross-linking strategy for fabricating vertically aligned conjugated polymers via cooperative hydrogen/halogen bonding and photopolymerization. The concept of integrating supramolecular pre-organization with covalent fixation is promising for nanomaterials design and applications. Historically, template-directed polymerizations that rely on preorganized vertical stacking have been extremely rare, because it is challenging to preserve delicate non-covalent order while forming covalent linkages along the stacking axis. The authors successfully overcome this dilemma, converting reversible supramolecular assemblies into stable one-dimensional covalent polymers under mild photochemical conditions. Therefore, it is a very nice work and appeals to the broad readership of Nature Communications, especially for polymer and supramolecular chemists. I thus think that this manuscript deserves to be published after addressing the issues described below."

Reply: We greatly appreciate the reviewer's positive comments on the work in this manuscript.

"1. In this templating polymerization protocol, self-assembly of the precursors driven by hydrogen/halogen bonding is very crucial for the subsequent photopolymerization. It is an important prerequisite for the ordered polymerization. However, in the manuscript, the investigation on the self-assembly behaves is not deep enough. It is recommended to do more studies on the self-assembly before photopolymerization, such as the measurement of binding stoichiometries and other characterizations (e.g. TEM or other electron microscope observations), which may provide more useful information to understand the templating mechanism."

Reply: We thank the reviewer for the insightful suggestion to strengthen the characterization of the self-assembly process prior to photopolymerisation. In the revised manuscript, we have incorporated two additional sets of experiments that directly address this point.

First, we quantified the binding stoichiometry between compound **1** and the pyridyl-containing monomer **2** by fluorescence titration. The resulting mole-ratio plots (revised Fig. 2a,b, Fig. S3) display a clear inflection at a 3:1 ratio, establishing that three molecules of **1** coordinate with one molecule of **2** in solution. This quantitative analysis complements our previous NMR evidence and provides a more rigorous understanding of the HB/XB-driven pre-organization.

Second, we added TEM measurements to visualize the supramolecular assemblies before photopolymerisation. As shown in the revised Fig. S4, the **1** + **2** mixture forms well-defined nanofibrous structures, whereas pure **1** does not generate any anisotropic morphology. These

observations are fully consistent with our AFM images, reinforcing the conclusion that cooperative binding between **1** and **2** is essential for forming the high-aspect-ratio columns that support vertical photopolymerisation.

Together, the added stoichiometric analysis and electron microscopy studies provide stronger evidence for the self-assembly mechanism.

“2. The overall English expression of the article is basically accurate, but there are some minor issues as follows.

In the title, the authors used “polymerisation”, however in the main section, the word “polymerization” appeared several times. I think both of them are OK, but it is better to choose one in a single paper.

On page 1, the abstract, the “align C3-symmetric monomers into ordered stacks” should be change into “C3-symmetric”.

At the end of page 3, “van der Waals contacts, or π - π interactions can builds supramolecular columns” should be “can build”.

On page 4, the authors said that “In optoelectronic devices, non-covalent vertical stacking facilitates uni- or multidimensional charge-transport channels and may reduce exciton recombination.” It is better to add some related references, making the sentence more convincing.

On page 6, the caption of Fig 1, it should be “The chemical structures of 1, 2, 3.” When they discussed the NMR spectra, they said “the pyridyl-H resonance (Fig. 1b)” and “the amide-NH protons of 1”. However, these protons already numbered as H3, H2 and H1 as shown in Fig 1. They can use these numbers when discussing them.

On page 9, they said that “without presence of 1 (Fig. 3, middle),” It is not very clear. They should divided Fig 3 into two parts, like Fig 3a and Fig 3b for the preparation of compound 6 and 5, respectively.”

Reply: We appreciate the reviewer’s careful reading of the manuscript and the helpful comments regarding minor issues in English usage. We have thoroughly revised the text according to the reviewer’s suggestions, including corrections to word choice, consistency in terminology, grammatical refinements, and improvements in clarity. The title, abstract, figure captions, and related sentences have all been updated to ensure uniform and accurate expression throughout the manuscript.

“3. There are some issues with the following pictures.

Figure 3 can be enlarged somehow. The structures in Figure 3 are too small and not clear enough. The structure of aggregate 4 appeared twice. This is not necessary. It can be combined with the preparation of polymer 6. And place the preparation of 5 in the lower part as Fig 3b.

The AFM image of compound 5 shown in Figure 5d is not clear enough and may cause confusion for the readers. It is suggested to replace it with a clearer one.”

Reply: We thank the reviewer for these helpful suggestions regarding the figure presentation and clarity. In the revised manuscript, we have carefully reworked the layout and appearance of the relevant figures. In addition, the AFM image of compound **5** in the original Figure 5d has been replaced with a clearer and more representative image, thereby avoiding potential confusion and improving visual comparability with the templated system. Overall, all figures in the main text have been adjusted for improved clarity, consistency, and readability in accordance with the reviewer’s comments.

“4. Since the characterization of AFM is crucial for understanding the morphology of the assemblies and vertically aligned conjugated polymer 6, I suggest adding a description of the AFM sample preparation in the SI section. This will facilitate researchers in the same field to follow.”

Reply: We thank the reviewer for highlighting the importance of providing detailed AFM sample preparation procedures. In the revised Supporting Information, we have added a dedicated section entitled “Procedure for AFM and TEM sample preparation”, which describes the preparation of AFM and TEM specimens in a step-by-step manner. Additionally, the treatment protocol for the silicon wafer used as the AFM substrate has been included under “The treatment of silicon wafer”. These additions ensure that the morphology studies can be reliably reproduced by other researchers in the field.

Response to Reviewer 2:

“In this manuscript, the authors present a conceptually elegant and technically rigorous study that reports a new synthetic strategy for achieving vertical covalent polymerisation of triphenylamine-based diacetylene assemblies through cooperative hydrogen-bonding (HB) and halogen-bonding (XB) interactions. This “self-assemble-then-cure” approach represents a conceptual advance with far-reaching interdisciplinary implications, providing a new synthetic platform that bridges supramolecular chemistry, polymer science, materials engineering, and optoelectronic applications. The integration of HB/XB cooperativity as a dynamic yet directional template not only achieves vertical polymerisation in solution—beyond crystalline topochemical constraints—but also opens a scalable route to conjugated nanostructures with high structural precision. Overall, this work is original, methodologically sound, and of clear significance to a broad readership spanning chemistry, physics, and materials science. However, before the manuscript can be accepted for publication, several minor revisions should be made to improve clarity and consistency.”

Reply: We thank the reviewer for their positive feedback and overall evaluation of the manuscript.

“1. Ensure all acronyms (HB, XB, PDA, MCH, etc.) are defined upon their first use in both the main text and supporting information.”

Reply: We thank the reviewer for pointing out the need to define all acronyms upon first use. In the revised manuscript, we have ensured that all abbreviations (including HB, XB, PDA, MCH, and others) are clearly defined at their initial appearance in both the main text and the Supporting Information. This revision improves clarity and readability for a broader audience.

“2. In Figure 4, specify the UV irradiation power density, wavelength, and apparatus geometry.”

Reply: We thank the reviewer for this helpful suggestion. In the revised manuscript, the UV irradiation conditions used in Figure 4, including the wavelength, power density, and irradiation geometry, have now been explicitly specified in the figure caption. These details clarify the photopolymerization conditions and improve the reproducibility of the experiments.

“3. In Figure 4, the four UV–vis and fluorescence spectra should be properly aligned for visual consistency. The panel labels (a), (b), (c), and (d) are currently misplaced and should be positioned uniformly at the upper left corner of each subfigure to comply with journal formatting standards.”

Reply: We thank the reviewer for the constructive comments on figure presentation. To improve visual clarity and consistency, we have revised the layout of Figure 4 by properly aligning the UV–vis and fluorescence spectra and repositioning the panel labels (a–d) uniformly at the upper left corner of each subfigure, in accordance with the journal’s formatting guidelines. In addition, for improved readability and logical flow, the order of some figures in the main text has been adjusted.

“4. In Figure 5, there are several layout inconsistencies that should be corrected in the revised version. Specifically, the border colors of panels (b) and (c) are not identical, and there is a visible white gap between panels (d) and (e). These formatting inconsistencies should be unified for a cleaner and more professional presentation.”

Reply: We thank the reviewer for pointing out these formatting issues in Figure 5. In the revised manuscript, the layout and appearance of all figures have been carefully adjusted, including unifying panel borders and correcting spacing inconsistencies. These revisions ensure a cleaner, more consistent, and more professional presentation throughout the main text.

“5. The procedures for gel preparation, photopolymerisation (photo-crosslinking) experiments, and the treatment after irradiation should be described in greater detail in the Supporting Information. Providing specific information on concentrations, irradiation time and intensity, and purification (e.g., washing or drying steps) would significantly improve the reproducibility and transparency of the experimental methodology.”

Reply: We thank the reviewer for highlighting the importance of providing detailed experimental procedures to improve reproducibility. In the revised Supporting Information, we have expanded the methodological descriptions accordingly. The gel preparation protocol has been included in a dedicated section entitled “Procedure for gel preparation.” In addition, detailed descriptions of the photopolymerisation (photo-crosslinking) experiments and post-irradiation treatment, including specific information on concentrations, irradiation time and intensity, as well as purification steps such as washing and drying, have been added to the section “General procedure for sample preparation and photopolymerization experiments.” These revisions substantially enhance the transparency and reproducibility of the experimental methodology, as suggested by the reviewer.

“6. The concept of achieving vertical photopolymerisation by pre-organising C₃-symmetric triphenylamine derivatives is intriguing. However, how can it be a general method and what are the potential applications? At least, the authors should discuss the potential generality and potential applications of this method in the manuscript, which would broaden its impact and inspire future studies in related fields.”

Reply: We thank the reviewer for this insightful comment regarding the generality and potential applications of the proposed strategy. In response, we have added a dedicated discussion in the Conclusion section to explicitly address this point. Specifically, we now note that, “beyond the triphenylamine system investigated in this work, preliminary supramolecular screening suggests that the HB/XB-templated vertical pre-organization is not limited to a single molecular framework and may be extendable to other pyridyl-containing monomers. While a comprehensive validation of molecular scope lies beyond the focus of the present study, this discussion highlights the broader conceptual relevance of the ‘self-assemble-then-cure’ strategy and its potential to serve as a general platform for constructing vertically aligned conjugated nanostructures”, thereby broadening the impact of the work and motivating future studies in related fields.

Response to Reviewer 3:

“This manuscript by Zhang, Li, and co-workers reports a supramolecular “self-assemble-then-cure” strategy to obtain triphenylamine-based poly(diacetylene) (PDA) nanofibers. Cooperative amide hydrogen bonding together with I···N halogen bonding pre-organizes a diyne monomer into 1D stacks, and short 254 nm UV irradiation triggers axial 2π+2π photopolymerization. After polymerization, the sacrificial template is removed by acid/base work-up, yet the nanofibrous morphology is retained. Evidence for templated assembly, photoconversion, and post-template integrity is provided by UV-vis/fluorescence spectroscopy, AFM imaging, and comparative 1H NMR analyses collected before and after templating/polymerization.

The assembly-photopolymerization-template-removal concept is attractive and the qualitative evidence is convincing. However, the characterization of the resulting polymer is not yet at the

level expected. Adding (i) at least one molecular-weight/DP handle, (ii) one vibrational or solid-state confirmation of PDA backbone formation/order, and (iii) clearer yield/conversion reporting would upgrade the work from a qualitative demonstration to a quantitatively characterized polymer material suitable for a broad materials audience.”

Reply: We thank the reviewer for the thoughtful and constructive evaluation of our work, and for highlighting the importance of strengthening the quantitative characterization of the resulting polymer materials. In response to these suggestions, we have revised the manuscript to address points (i)–(iii) as follows.

For (i), we introduced a molecular-size handle by performing statistical length analysis on several hundred PDA nanofibers based on SEM images of the final polymer. Using a conservative repeat distance for the poly(diacetylene) backbone, this analysis enables estimation of an apparent axial degree of polymerization (DP), thereby providing a quantitative descriptor of effective chain growth in this templated system.

For (ii), we added FTIR spectroscopy to provide vibrational confirmation of diacetylene-to-PDA backbone formation. The disappearance of the characteristic butadiyne $\nu(\text{C}\equiv\text{C})$ band and the emergence of bands associated with conjugated $\text{C}=\text{C}$ stretching offer direct vibrational evidence for successful PDA formation, complementing the optical spectroscopic data.

For (iii), we improved conversion reporting (approximately 96%) by conducting a semi-quantitative analysis of the residual diacetylene signal in the FTIR spectra. Integration of the $\nu(\text{C}\equiv\text{C})$ band before and after irradiation enables an approximate estimation of the conversion efficiency, providing an additional and independent handle on the extent of photopolymerization.

“A major weakness is the absence of standard polymer metrics ($M_n/M_w/D$). At present the study establishes that “a polymer is formed,” but it does not show how large or how disperse it is. The authors are encouraged to consider one or more of the following routes:

**Controlled scission to generate soluble fragments, followed by SEC-MALS to obtain $M_n/M_w/D$;*

**An end-group/chain-stopper strategy (low-% chromophoric stopper) combined with $q\text{NMR}$ or absorbance ratios to estimate DP;*

**AFM/TEM length statistics on a large number of fibers, converted to DP using a conservative repeat distance.”*

Reply: We thank the reviewer for emphasizing the importance of quantitative polymer metrics. In response, we adopted the third approach suggested by the reviewer and performed statistical length analysis on several hundred nanofibers based on SEM images of the final polymer (now shown in Fig. 5h and Fig. 5i). Using a conservative repeat distance for the poly(diacetylene) backbone, this analysis allows us to estimate an apparent axial degree of polymerization (DP), providing a quantitative measure of the effective covalent chain growth achieved in the templated system.

The first two approaches, although conceptually powerful, are not readily applicable to the present system. We did attempt SEC-MALS measurements on the final PDA products directly, because of their acceptable solubility. However, these experiments consistently yielded anomalously low M_n and M_w values which even lower than compound **2**, indicating that the obtained values are not physically meaningful. We attribute this behavior to the intrinsic nature of the PDA materials, which form rigid, rod-like one-dimensional nanofibers that exist as bundled aggregates rather than discrete molecular chains in solution.

Additionally, implementation of an end-group or chain-stopper strategy would require substantial redesign of the monomer structure and reaction conditions, which lies beyond the scope of the current study and would fundamentally alter the templated photopolymerization process under investigation.

While we acknowledge that AFM/SEM-based length statistics are not equivalent to conventional $M_n/M_w/\bar{D}$ values, we emphasize that this approach provides a meaningful and experimentally accessible quantitative descriptor for poorly soluble or topochemically polymerized systems. The derived DP values demonstrate that the templated reaction yields extended covalent backbones over hundreds of nanometres, thereby moving the system well beyond oligomeric regimes and directly addressing the reviewer's concern regarding polymer size.

“In addition, backbone verification by vibrational spectroscopy is missing. The authors should include Raman and/or IR data diagnostic for diacetylene \rightarrow PDA conversion ($C\equiv C$ loss, $C=C$ growth), and—if possible—quantify the conversion by band-area analysis to complement UV-vis kinetics.”

Reply: We thank the reviewer for this helpful suggestion. In the revised manuscript, we have added FTIR spectra (Fig. 5c) to provide vibrational spectroscopic verification of the diacetylene-to-PDA conversion. The FTIR data clearly show the attenuation of the diacetylene $\nu(C\equiv C)$ band at $\sim 2210\text{ cm}^{-1}$ together with the emergence of conjugated $C=C$ stretching bands characteristic of PDA backbones, confirming successful backbone formation. In addition, by comparing the band areas of the $\nu(C\equiv C)$ absorption before and after irradiation, we obtained a semi-quantitative estimate of the conversion, which is approximately 96% for the templated system. These results provide direct vibrational evidence for efficient diacetylene-to-PDA conversion and complement the UV-vis spectroscopic analysis presented in the main text.

“One solid-state method (ssNMR for backbone signals or PXRD for topochemical order) would further support claims of an ordered 1D conjugated structure.”

Reply: We appreciate the reviewer's suggestion to include solid-state techniques such as ssNMR or PXRD to further support structural features of the system. At the present stage, however, the PDA nanofibers cannot be readily prepared in the relatively large quantities typically required for

solid-state NMR or PXRD measurements. More importantly, we believe that the key structural information targeted by these techniques is already sufficiently addressed through the combination of complementary methods employed in this work.

Specifically, the proton NMR in solution proves the occurrence of template-directed polymerization and FTIR spectroscopy provides direct vibrational evidence for the conversion of diacetylene units into a covalent PDA backbone, while UV–vis and fluorescence spectroscopy confirm the establishment of extended conjugation. Microscopic analyses (AFM and SEM), together with statistical length analysis of several hundred nanofibers, directly visualize and quantitatively describe the one-dimensional morphological characteristics and axial continuity of the resulting covalent framework. Collectively, these data sets offer a coherent and mutually reinforcing picture of a structurally coherent one-dimensional conjugated architecture formed via templated photopolymerisation. In this context, we consider that additional ssNMR or PXRD measurements would likely provide limited incremental insight relative to the substantial experimental cost and material requirements.

“It would also help to report elemental analysis (e.g., halogen/N content before and after work-up) and/or provide simple kinetic fitting of the spectral evolution to estimate the degree of conversion/cross-linking and to demonstrate complete template removal.”

Reply: We thank the reviewer for this constructive suggestion regarding elemental analysis to further substantiate template removal. We did attempt to probe the elemental composition of the final materials using SEM-based energy-dispersive X-ray spectroscopy (EDS). However, due to the intrinsically nanoscale dimensions of the PDA nanofibers, both in thickness and overall mass, reliable elemental signals from the polymer could not be unambiguously resolved. In practice, the weak signals from light elements (C, N) and residual halogen species were largely overwhelmed by the strong background contribution from the silicon substrate, rendering quantitative elemental analysis inconclusive.

As an alternative and complementary line of evidence, comparative ^1H NMR analysis provides strong support for effective removal of template molecules. As shown by comparing the spectra of the assembly and the final polymer product (Fig. 1 and 5b), resonances attributable to the aromatic protons of template **1** are not discernible in the spectrum of polymer **8**. Given that residual template molecules, if present in significant amounts, would be expected to contribute detectable aromatic signals, their absence indicates that compound **1** is not retained in the final material to any appreciable extent. Together, these observations support the conclusion that the acid/base work-up enables effective removal of the supramolecular template. A corresponding description has also been added in the main text on Page 13.

“Overall, the core narrative—from self-assembled axial photopolymerization to template removal—is compelling and already well supported by spectroscopy and AFM. The manuscript will be substantially strengthened by the above quantitative additions, which will move it from “polymer formed” to a fully and rigorously characterized polymer material.”

Reply: We are grateful to the reviewer for the positive and encouraging comments on this work. We hope that the revisions are satisfactory.

Response to Reviewer 4:

“I co-reviewed this manuscript with one of the reviewers who provided the listed reports. This is part of the Nature Communications initiative to facilitate training in peer review and to provide appropriate recognition for Early Career Researchers who co-review manuscripts.”

Reply: We really thank the co-reviewer for the thoughtful comments on this manuscript.

Again, we appreciate the Reviewers’ comments which have greatly improved our manuscript. With these changes and responses, we hope that the manuscript now meets the standards for acceptance as an Article in *Nature Communications*.

Sincerely yours,

Shijun Li